# Melanoma Management: From Epidemiology to Treatment and Latest Advances

**DOI:** 10.3390/cancers14194652

**Published:** 2022-09-24

**Authors:** Joana Lopes, Cecília M. P. Rodrigues, Maria Manuela Gaspar, Catarina Pinto Reis

**Affiliations:** 1Research Institute for Medicines, iMed.ULisboa—Faculty of Pharmacy, Universidade de Lisboa, Av. Professor Gama Pinto, 1649-003 Lisboa, Portugal; 2Instituto de Biofísica e Engenharia Biomédica, IBEB, Faculdade de Ciências, Universidade de Lisboa, Campo Grande, 1749-016 Lisboa, Portugal

**Keywords:** melanoma, epidemiology, etiology, diagnosis, treatment, nanotechnology, clinical trials

## Abstract

**Simple Summary:**

Melanoma is a major public health issue that claims the lives of thousands of people every year. Furthermore, the outlook for the coming years is not encouraging with increasing morbidity and mortality trends. This review aims to offer an updated overview of various aspects related to cutaneous melanoma, from epidemiology, etiology, clinical presentation, prevention, diagnosis and staging. Moreover, conventional treatments currently available as well as the latest advances in clinical trials regarding new drugs and/or combinations, including nanotechnology-based strategies are also reviewed.

**Abstract:**

Melanoma is the deadliest skin cancer, whose morbidity and mortality indicators show an increasing trend worldwide. In addition to its great heterogeneity, melanoma has a high metastatic potential, resulting in very limited response to therapies currently available, which were restricted to surgery, radiotherapy and chemotherapy for many years. Advances in knowledge about the pathophysiological mechanisms of the disease have allowed the development of new therapeutic classes, such as immune checkpoint and small molecule kinase inhibitors. However, despite the incontestable progress in the quality of life and survival rates of the patients, effectiveness is still far from desired. Some adverse side effects and resistance mechanisms are the main barriers. Thus, the search for better options has resulted in many clinical trials that are now investigating new drugs and/or combinations. The low water solubility of drugs, low stability and rapid metabolism limit the clinical potential and therapeutic use of some compounds. Thus, the research of nanotechnology-based strategies is being explored as the basis for the broad application of different types of nanosystems in the treatment of melanoma. Future development focus on challenges understanding the mechanisms that make these nanosystems more effective.

## 1. Introduction

Skin is the largest organ of the human body, covering about 2 m^2^ of surface and weighing approximately 3.6 kg in adulthood. It has important functions that guarantee protection against physical, chemical and biological agents, regulation of body temperature and production of antimicrobial compounds to prevent infections [1,2]. Skin is divided into three layers, from the outermost to the innermost: epidermis, consisting mainly of keratinocytes (95%) and some dendritic cells such as melanocytes, Merkle and Langerhans cells; dermis, composed of connective tissue, essentially collagen and elastic fibers, as well as blood vessels, nerve endings and glands, such as sebaceous and sweating; and hypodermis, whose function is the connection between the dermis and the underlying organs [3,4].

Skin diseases affect millions of people around the world and cover a wide range of problems going from chronic diseases such as atopic dermatitis and psoriasis to neoplasms. Although potentially treatable, the latter is associated with a high mortality rate [5,6,7]. Regarding skin cancers and according to the International Agency for Research on Cancer (IARC), in 2020, there were around 1.52 million new cases and 121,000 related deaths [8,9]. Every 4 min, one person dies from skin cancer [10]. Moreover, the prospects for the next 20 years by the same agency are not optimistic as increases of 78 and 73% in melanoma’s incidence and mortality, respectively, are expected [8,9].

Neoplastic changes can occur in diverse skin layers, namely, at the epidermal layer. Epidermal cells’ malignancies can be divided into melanoma, which will be the subject of this work, and non-melanoma skin cancers depending on whether the cells of origin are melanocytes or keratinocytes and Merkel cells, respectively [11,12].

Melanoma is a word derived from the Greek *melas* “dark” and *oma* “tumor”, having been firstly described by Hippocrates in the 5th century BC [13]. It is an aggressive disease that derives from the malignant transformation and uncontrolled proliferation of melanocytes [14,15]. Melanocytes derive from an existing structure in vertebrate embryos denominated neural crest, and their main function is the production of melanin (melanogenesis) and its storage until transferred to keratinocytes [16,17,18]. Melanin is an important pigment responsible for the color of the skin, hair and eyes. It plays an important role in protecting skin from sun exposure, neutralizing reactive oxygen species (ROS) and storing ions [19]. As already stated, melanocytes are mostly present in the epidermis, more precisely, in the innermost layer out of four layers, *stratum basale* [20]. The malignant transformation of these epidermal melanocytes comprises cutaneous melanoma, independently of being metastatic or non-metastatic.

Cutaneous melanoma is the focus of this review. This malignancy includes the vast majority of diagnosed cases [21,22] and its principal subtypes are superficial spreading melanoma, nodular melanoma, *lentigo maligna* melanoma and acral lentiginous melanoma. However, melanocytes can also be present in other tissues, such as hair follicle bulbs, eyes, inner ear, mucosa and central nervous system, comprising in these cases the non-cutaneous form [21,22,23,24]. Furthermore, although melanoma represents the least common form of skin cancer, it is the most aggressive and lethal [15,25,26]. In addition to the considerable social impact, the costs and complexity of health care in advanced stages have attracted attention. In the United States of America, for example, the costs associated with melanoma represent about 40% of the annual budget allocated to skin cancer [14]. Contrarily, despite having 18–20 times higher incidence, non-melanoma skin cancers, whose principal forms are Basal Cell Carcinoma (BCC) and Squamous Cell Carcinoma (SCC), present slow growing and low tendency to metastasize and thus are rarely lethal [27,28,29,30].

In terms of treatment, the management of melanoma has rapidly developed in recent years. The landscape of available therapeutic options was limited to surgery, radiotherapy and chemotherapy until 2011 [31,32]. From then on and leveraged with the approval of the first immune checkpoint inhibitor, ipilimumab, and the first small-molecule kinase inhibitor, vemurafenib, approvals followed one another. However, despite the unquestionable progress, these therapies remain limited in some cases, with resistance and the incidence of adverse side effects [32,33]. Thus, alternative options continue to be investigated either in a clinical or pre-clinical phase [34,35,36,37,38].

Nanotechnology, a concept created in the middle of the last century, has also proved to be a valuable and promising tool [39], with several lipidic, polymeric, metallic and hybrid nanosystems already approved in various areas, such as cancer [40,41,42]. The improvement of physicochemical properties of the compounds, such as water solubility and stability and consequently the associated pharmacokinetic and pharmacodynamic profiles, as well as the possibility of targeting these nanosystems to the tumor site, either passively or actively, might change the current therapeutic strategy for melanoma [43,44]. Thus, this review aims to provide an updated overview of the various aspects of cutaneous melanoma, starting from epidemiology, etiology, clinical presentation, prevention, diagnosis, staging, available conventional treatments and ending with the latest advances in clinical trials with new drug combinations but also including nanotechnological approaches (Figure 1).

## 2. Cutaneous Melanoma

### 2.1. Epidemiology and Etiology

Melanoma is a type of malignancy whose worldwide incidence rate has been increasing very rapidly over the last decades [14,45]. Light-skinned populations such as those in North America, Northern Europe, Australia or New Zealand have seen the annual incidence of melanoma increase by 4–6% [14]. According to IARC estimates, there were around 325,000 new cases associated with cutaneous melanoma in 2020 [46]. Representing 1.7% of all cancer diagnoses [47], melanoma is ranked as one of the most common cancers worldwide, probably reaching 57,000 deaths in the same period [48]. Figure 2 briefly illustrates the incidence and mortality distribution in different countries. There are several factors contributing to this scenario, namely, atmospheric ozone depletion, global warming and air pollution [49,50]. The same agency predicted until 2040 an increase of about 57 and 68% in the number of new cases and related deaths, respectively [51,52].

The transformation of melanocytes into malignant cells is a really complex process that results from the interaction of different modifiable and non-modifiable risk factors (Table 1) [21]. Exposure to UV radiation from sunlight or the use of tanning devices are considered the strongest modifiable risk factor of melanoma, responsible for about 60–70% of all diagnosed cases [53,54]. The IARC even classified them as carcinogens (Group 1), alongside tobacco and asbestos [55,56]. It is not surprising, therefore, that it is in the equatorial regions, where the hours of sunlight exposure are higher, that the highest rates of melanoma incidence are verified [22]. In 2020, the highest incidence rates of melanoma reported by IARC occurred precisely in Australia and New Zealand, with an age-standardized rate (ASR) per 100,000 habitants of 36.6 and 31.6, respectively. In contrast, the following countries with a great incidence of ASR were all located at higher latitudes, as is the case of Denmark, The Netherlands, Norway, Sweden, Switzerland and Germany [57]. A lower incidence was observed in the southern countries of Europe, and this can be explained by the differences in skin pigmentation of the populations of these regions, reflecting the patterns of incidence according to the ethnicity reported further on. The prevalence of darker pigmentation also explains the low values of ASR in other populations living close to the equator, as is the case in many Asian and African countries [14,57]. In addition, altitude has also been suggested as a risk factor, since changes in ozone absorption, less cloud cover as well as an increase in surface reflectance in areas with snowfall, can increase the exposure to UV radiation [14]. Approximately 6.8% of solar radiation belongs to the UV spectrum, which comprises three types of radiation: UVC, UVB and UVA (wavelength range: 200–280, 280–315 and 315–400 nm, respectively) [58]. UVC radiation, as well as almost all UVB radiation, are absorbed by the stratospheric ozone layer, leading that only 5% of UVB radiation and 95% of UVA radiation reaching the earth’s surface [59,60]. Thus, it is UVA, predominantly, and UVB radiation that are responsible for the mutagenic and proliferation effects in melanin-producing cells. UVB radiation, in addition to direct DNA damage, also promotes an inflammatory response that stimulates angiogenesis and the survival, proliferation and metastatic potential of mutated cells. In turn, UVA radiation indirectly triggers DNA strand damage, lipid peroxidation and protein damage through oxidative stress [53,54,58,61].

Although melanoma essentially affects an older population, it is clear that there is an increase in prevalence in a younger population (Figure 3) [14]. In this sense, while colon and lung cancer have an average diagnosis of 68 and 70 years old, respectively, melanoma, is only 57 years [21,22].

Furthermore, sex is also an influential factor. At younger ages, women have the highest incidence rate of melanoma, but as age increases, the trend is reversed, and men are the most predisposed [21,22]. However, age aside, the overall incidence of melanoma is clearly higher in men than in women, with an ASR of 3.8 and 3.0 for 100,000, respectively [47,63,64]. The reason for this sex disparity is still not completely clear. However, there is evidence that in addition to the contribution of sex-specific behavioral factors, there are also intrinsic biological differences. An example is female genetic heterogeneity derived from the epigenetic inactivation of one of the two X chromosomes [64].

Regarding ethnicity, the incidence of melanoma varies considerably. Caucasians are more likely to develop melanoma compared to dark-skinned populations (2.4 vs. 0.1%) [45,65]. Skin color is the reason for this differentiable risk, being influenced by several factors such as the mixture of carotenoids, oxy-/deoxy-hemoglobin, and most outstandingly, different types of melanin. Furthermore, the melanogenic activity as well as the number, size and packaging of melanosomes impact skin color [60,66]. There are two different forms of melanin present in the human body: eumelanin and pheomelanin. Eumelanin is a brownish-black pigment, mainly present in dark-skinned populations, which exerts a protective action on the skin, spreading and absorbing part of the UV radiation as well as eliminating free radicals. In contrast, the own production mechanism of pheomelanin, a reddish-yellow pigment mainly produced by light-skinned people, promotes oxidative stress, leading to greater susceptibility of melanocytes to DNA damage [60,67,68,69,70].

Finally, being melanoma associated with a great genetic component, several genetic alterations (hereditary and somatic) have been reported over the last few years [71]. Germline mutations account for about 5–10% of cases, and among these, the cyclin-dependent kinase inhibitor 2A (CDKN2A) mutation is the most prevalent. However, mutations in several other genes such as melanocortin 1 receptor (MC1R), microphthalmia-associated transcription factor (MITF), cyclin-dependent kinase 4 (CDK4), protection of telomeres 1 (POT1), telomerase reverse transcriptase (TERT), adrenocortical dysplasia (ACD), telomeric repeat-binding factor 2-interacting protein 1 (TERF2IP) and BRCA1-Associated Protein 1 (BAP1) are also present in melanoma-prone families [72,73]. On the other hand, somatic mutations are mostly associated with mitogen-activated protein kinase cascade. They are classified according to the most frequently mutated genes into four principal types: BRAF (serine/threonine protein kinase B-raf), the most frequent, NRAS (neuroblastoma RAS viral oncogene homolog), NF1 (neurofibromin 1) and triple wild type (no mutation in any of these three genes) mutations [71,74,75]. Moreover, also specific genetic conditions such as albinism and xeroderma pigmentosum increase melanoma’s risk [76,77].

**Table 1 cancers-14-04652-t001:** Modifiable and non-modifiable risk factors of melanoma [15,21,76,77,78].

**Modifiable risk factors**	Exposure to UV radiation (e.g., sunlight or use of tanning devices)
	History of blistering sunburns at a young age
	Medications (e.g., psoralen or immunosuppressive drugs)
	Environmental exposure to chemicals (e.g., heavy metals or pesticides)
**Non-modifiable risk factors**	Age
	Sex
	Ethnicity
	Individual phenotypic characteristics (e.g., skin and light eyes, red or blond hair and high density of freckles)
	Clinical characteristics of the patient (e.g., increased number of common nevi or presence of atypical nevi)
	Personal and family history of skin cancers
	Personal history of diseases that compromise the immune system (e.g., hematologic malignancies or infection by HIV)
	Genetic alterations
	Specific genetic conditions (e.g., albinism or xeroderma pigmentosum)

### 2.2. Clinical Presentation

Melanoma is one of the most heterogeneous cancers, both in terms of etiology as previously mentioned and in terms of its clinical characteristics. These characteristics are dependent on several factors such as the origin and anatomical location [79,80].

There are four main types of cutaneous melanoma: superficial spreading melanoma, nodular melanoma, *lentigo maligna* melanoma and acral lentiginous melanoma, each one of them associated with different epidemiological, dermatological and histopathological factors [24,81].

Briefly, superficial spreading melanoma, the most diagnosed type of melanoma (70–80% of the total cases), is the most prevalent in the population between 30 and 50 years old [82]. Usually, it appears as a macula that tends to evolve into a palpable papule or nodule with irregular margins and a broad range of colors. Histologically, it is common to observe a pagetoid and nested spread of malignant melanocytes in intraepidermal tissues. Intradermal nests of melanocytic proliferation are also observed [24,83]. Although it can arise in any localization, it is more frequently found on the trunk in men and on the lower extremities in women. Intraepidermal horizontal growth can last for a long time and only then become invasive [82,84].

Nodular melanoma is the second type of melanoma more frequent (15–30% of cases). It can also appear in any location; however, it is more common in the trunk, head and neck. This type of cancer is the most aggressive [82]. The brown or black or even blue-black lesions of this type of tumor have a wide range of presentations. It can appear as a polypoid exophytic tumor, a prominent plaque with irregular shapes, or a smoothly surfaced cutaneous nodule. Histologically, intradermal nests and aggregates of tumor cells are observed, while the intraepidermal melanocytic proliferation is minimal and overlying the dermal tumor [83].

Lentigo maligna melanoma represents 5 to 15% of cases. It is more frequent on the sun-damaged faces of elderly people, mainly on the nose and cheeks. This type of tumor has an antecedent skin lesion called lentigo maligna, which can have a horizontal growth period of many years before it becomes invasive [82]. Typically, it might occur as a tan, brown and black flat tumor of irregular contours, presenting flecks with the same colors. Histologically, an extensive melanocytic proliferation is observed in the dermal-epidermal junction that extends through the hair follicle epithelium. Intradermal nested of epithelioid or spindled cells are also observed [83].

Lastly, with only 2–8% of the cases, usually found in people of color, there is acral lentiginous melanoma. It arises mainly on the palms of the hands and soles of the feet as an atypical pigmented macule that develops an elevated plaque or nodule [81,82]. The histological feature is an extensive and poorly circumscribed lentiginous growth of nested tumor cells parallel to the epidermis [85]. Unfortunately, it is detected at advanced stages [86].

Additionally, other variants more rarely reported of cutaneous melanoma comprise the desmoplastic, polypoid, primary dermal, verrucous and amelanotic melanomas [24,87].

### 2.3. Prevention, Diagnosis and Staging

As an important public health problem, the strategy to tackle melanoma must be based on two essential pillars: prevention and early diagnosis [45,88].

Prevention strategies fundamentally focus on promoting the health literacy of the population, looking for behavioral changes regarding modifiable risk factors as well as raising awareness of the importance of sun protection habits. In addition, there has been increasing interest in the close surveillance of individuals with high-risk characteristics for developing melanoma [54,89,90,91]. On the other hand, an early and accurate diagnosis of the disease is essential to improve the outcomes. The overall survival in 5 years decreases from 99% when detected in the stage IA melanoma to 15–20% in the most advanced stage (IV) [76].

The starting point for clinical diagnosis is the visual screening of suspicious lesions, greatly supported by the ABCDE rule (Asymmetry, Border irregularity, Color heterogeneity, Diameter ≥6 mm and Evolving) as schematized in Figure 4 [92,93]. In the case of nodular melanoma, the mnemonic rule is EFG (Elevated, Firm and Growing). In addition, a comparative dermoscopic analysis must be carried out [26,94]. Moreover, an excisional, whenever possible, punch or shave biopsy with subsequent histopathological analysis should be performed. The histopathological analysis allows the identification of different parameters, such as histotype of melanoma, Breslow thickness, Clark level, mitotic index and the presence or absence of ulceration [95,96,97]. Furthermore, immunohistochemical analysis to identify melanocytic (S-100 protein, Melan-A, HMB-45 or SOX10) and proliferation markers (MIB-1), lymph node biopsy, genetic characterization of targetable somatic mutations and measurement of LDH serum levels can also be performed in specific cases [26,98,99,100]. In addition, many non-invasive pre-biopsy imaging technologies have been developed in recent years. Whole-body 3-D imaging, reflectance confocal microscopy, optical coherence tomography, high-frequency ultrasound and multispectral digital skin lesion tools such as MelaFind, approved by FDA in 2011, are examples [101,102].

Furthermore, accurate staging of the tumor is also a crucial step. This will allow the clinician to assess the prognosis of the patient and select the best and most appropriate therapeutic strategy [103]. The most used cutaneous melanoma staging system belongs to the American Joint Committee on Cancer (AJCC). This system, called TNM staging system, is based on an assessment of three aspects, each one with more than one criterium: (T) Breslow tumor thickness of the primary tumor and presence or not of ulceration; (N) number of involved lymph nodes and the presence or not of in-transit, satellite and/or microsatellite metastasis; and (M) anatomic site of distant metastasis and LDH levels. After analysis, patients are included in specific pathological stages groups, which are divided as follows: 0, I (IA and IB), II (IIA, IIB and IIC), III (IIIA, IIIB, IIIC and IIID) and IV. Briefly, while in stage 0 cancer cells are limited to the epidermis, in stages I and II tumors are classified according to their thickness and degree of ulceration. In turn, stages III and IV classification are attributed as soon as the involvement of lymphatic tissues occurs and when there is dissemination to one or more vital organs, respectively [104].

## 3. Challenges and Opportunities for Cutaneous Melanoma Treatment

As previously mentioned, in the early stages of the disease, the prognosis might be positive, and the patient can be successfully treated surgically [105]. However, as the disease progresses, survival rates significantly decrease [76]. Until 2010, only radiotherapy and chemotherapy were considered alternatives to surgery [31,100,106]. Later on, the growing knowledge about the pathogenesis of the disease, the role of the immune system and the greater capacity for genomic sequencing, allowed the identification of new targets [106,107,108]. Thus, marketing authorization of different drugs and/or combinations has taken place [107,108,109]. However, intra and intertumoral heterogeneity continues to overlap, limiting therapeutic success [107,110,111], but according to the ClinicalTrials.gov database, a large number of clinical trials have been carried out testing new therapeutic strategies, including those based on nanotechnology.

### 3.1. Current Available Strategies

The currently available strategies for the treatment of melanoma are surgery, radiotherapy, chemotherapy, immunotherapy and targeted therapy (Table 2). The selection of the most suitable therapeutic strategy depends not only on the anatomic location, stage and genetic profile of the tumor but also on the age and general health status of the patient [32]. The different therapeutic alternatives will be discussed in the following Sections.

#### 3.1.1. Surgery Resection

Whenever possible, surgical removal with adequate margins is the first-line treatment of melanoma [105]. Although it is mainly applied in patients up to stage II melanoma, it is also often an option for stage III patients [114,115] or even when the disease has already metastasized to other organs (stage IV) [116,117,118,119,120]. However, especially in some patients at stages II and patients at stages III and IV, surgery alone has limited curative potential. Thus, radiotherapy, chemotherapy, immunotherapy or targeted therapy are often used as adjuvant treatments [116,121,122].

#### 3.1.2. Radiotherapy

Melanoma is a relatively radioresistant tumor as it has the ability to effectively repair DNA damage caused by radiation [123]. Therefore, the choice of radiotherapy as a first-line treatment is applied for exceptional cases, for example, given the impossibility of performing surgery or as a complement to some situations where there is a high risk of recurrence. On the other hand, radiotherapy is widely used as a palliative treatment for metastatic melanoma (stage IV). New techniques such as stereotactic radiosurgery and stereotactic body radiotherapy have commonly been used in the treatment of brain, lung or liver metastases. Compared with whole-brain radiotherapy in the case of treatment of brain metastases, promising results and less severe adverse side effects have been observed [116,124,125]. Nonetheless, the combination of radiotherapy with systemic therapeutic options is currently under study in some clinical trials (NCT02858869 and NCT04902040) [126].

#### 3.1.3. Chemotherapy

Although chemotherapy remains a therapeutic option for melanoma management, especially in palliative or relapsed situations, in metastatic advanced stages of the disease, new therapeutic choices are preferred [127].

The main disadvantages associated with chemotherapy are the lack of specificity for tumor cells and consequent low drug accumulation at the tumor microenvironment. Thus, therapeutic benefits are limited and the incidence of adverse side effects is prominent [43,123,127]. To the best of our knowledge, the DNA alkylating agent dacarbazine (DTIC) remains the only drug approved both by FDA and EMA [128]. Generally, in the various clinical trials conducted, DTIC response rate was around 10 to 20%, with most responses being partial and not sustained over time. In addition, nausea, vomiting and myelosuppression are the most common adverse side effects [127,129].

In addition to DTIC and despite not being officially approved, many other chemotherapeutic agents have been used off-label, namely, temozolomide (TMZ), nitrosoureas, paclitaxel, docetaxel and cis/carboplatin [100,128].

TMZ is a DTIC analog approved for glioblastoma but frequently used in metastatic melanoma [127]. A phase III clinical trial comprising 305 volunteers with advanced disease found similar efficacy between both drugs (13.5 vs. 12.1% objective response rate for TMZ and DTIC, respectively) [130]. Besides this optimistic result, the oral route of administration and the ability of TMZ to cross the blood–brain barrier, relevant in the case of brain metastases, are other advantages [127,131,132].

Like the previous ones, nitrosoureas, such as photomustine, carmustine and lomustine, are alkylating agents, generally used in combination with other drugs [127]. The anticancer activity of these compounds is identical to the gold standard, DTIC. However, they present a much unfavorable toxicity profile [129].

In turn, taxanes, such as paclitaxel and docetaxel, and platinum compounds, namely, carboplatin and cisplatin, have also been clinically tested in the treatment of melanoma as monotherapy. However, because of their moderate anticancer activity, the combination of these and other therapeutic classes continues to be investigated [127,129,131].

#### 3.1.4. Immunotherapy

Immunotherapy has as its primordial objective the stimulation and activation of the immune system. It started with the FDA approval of interleukin 2 (IL-2) and interferon alfa-2b in the 1990s, with the latter being also approved by EMA. Since 2011, there has been an important shift in the role of immunotherapy in melanoma like immunological checkpoint inhibitors (ICIs) [133,134]. Interestingly, melanoma was the first malignancy to take advantage of ICIs [135]. There are four classes of ICIs monoclonal antibodies approved for the treatment of melanoma, namely, ipilimumab, an antagonist of cytotoxic-T lymphocytes antigen 4 (CTLA-4); nivolumab and pembrolizumab, antagonists of programmed cell death protein 1 (PD-1); atezolizumab, an antagonist of programmed cell death ligand 1 (PD-L1) [106]; and more recently, in 2022, relatlimab-rmbw, an antagonist of lymphocyte activation gene-3 (LAG-3) [136]. These last two drugs are already approved for melanoma indication by FDA but not by EMA.

CTLA-4, PD1, PD-L1 and LAG-3 are immune checkpoint proteins expressed on T cells membrane and involved in the signaling pathways that lead to its suppression. T cells are physiologically essential in the maintenance of immune tolerance. However, these immunological checkpoints are often used by cancer cells for immune evasion by down-regulation of their antitumor responses. ICIs, by selectively binding to these proteins, allow overcoming tumor-induced inhibition of T cell functions, re-establishing the anti-tumor immune responses [137,138,139,140,141,142,143].

Nonetheless, there is still a large percentage of patients who present innate and acquired resistance, not responding to this type of therapy [144,145,146,147]. Moreover, due to their mechanism of action that stimulates the immune system, ICIs are associated with an imbalance in immune tolerance leading to numerous immune-related adverse events (irAEs). The most common are dermatological, such as skin rash, pruritus and vitiligo; gastrointestinal such as diarrhea and colitis; hepatic; and endocrine conditions, such as hyperthyroidism and hypothyroidism [138,146,148].

Another relatively recent immunotherapeutic strategy approved for the treatment of advanced melanoma is talimogene laherparepvec (T-VEC or Imlygic^®^), the first oncolytic virus approved in the cancer treatment field. T-VEC is a herpes simplex virus type 1 genetically modified to reduce pathogenicity and increase immunogenicity. The mechanism of action not only involves the infection and death of melanoma cells but also the local and systemic stimulation of immune response. The main adverse side effects associated are fatigue, chills and fever, which tend to diminish with consecutive administrations [149,150].

#### 3.1.5. Targeted Therapy

Cutaneous melanoma has a high rate of genetic alterations, with 7 out of 10 diagnosed cases having mutations in genes of the main signaling pathways. These oncogenic mutations promote the activation of cell signaling pathways, leading to the proliferation of malignant cells without any type of control [151,152]. The aim of target therapy is precisely to stop this widespread proliferation through the inhibition of the mutated genes [106,138]. The mitogen-activated protein kinase (MAPK) cascade is the most frequently mutated in melanoma, being composed of a receptor tyrosine kinase as well as RAF, RAS, MEK and ERK proteins [153]. Mutations in the BRAF, NRAS and NF1 genes are the most frequent and occur in approximately 50, 25 and 14% of melanoma cases, respectively [154,155]. As previously mentioned, there are already several molecules of this therapeutic class used for clinical management of melanoma, such as vemurafenib, dabrafenib and encorafenib (BRAF inhibitors), and trametinib, cobimetinib and binimetinib (MEK inhibitors). Although they are distinct between different BRAF and MEK inhibitors, arthralgia; fatigue; diarrhea; pyrexia; photosensitivity; and skin, cardiovascular and ocular toxicity are the main adverse effects associated with this type of therapy [155,156]. Finally, the main limitation inherent in this type of treatment is related to the development of resistance [32,154,156,157].

#### 3.1.6. Combination of Therapeutic Approaches

As previously reported, immunotherapy and targeted therapy have shown good results in terms of efficacy; however, they are associated with some limitations. Thus, taking advantage of the targeted and immunotherapies, the American and European guidelines recommend a combination of therapeutic approaches [133,158,159,160]. There are already several combinations approved for clinical use (Table 3). One of the most recently approved combinations was nivolumab with relatlimab-rmbw (*Opdualag*) on 18 March 2022, by the FDA, for patients with unresectable or metastatic melanoma from 12 years of age [136].

### 3.2. Ongoing Clinical Trials

As described so far, the last few years have been marked by tremendous efforts regarding the development of new therapeutic strategies for the treatment of melanoma [161]. Nevertheless, and as a result of the highly aggressive and heterogeneous nature of this malignancy, existing therapies remain limited [100]. Hereupon, new drugs and/or combinations are being tested to find more and better therapeutic options as described in Table 4. According to the ClinicalTrials.gov database, there are currently 853 clinical trials worldwide focusing on melanoma as an object of study [162]. Our research was restricted to ongoing interventional studies, listed as “not yet recruiting”, “recruiting”, “enrolling by invitation” and “active, not recruiting”.

### 3.3. Completed and Undergoing Innovative Nanotechnological Approaches on Clinical Trials

Nanotechnology-based strategies can play a key role in improving drug efficacy. The controlled delivery and targeting of loaded drugs at the cellular level overcoming the biological barriers is one of the added value of this type of option [43,44]. Currently, several types of nanosystems namely liposomes, solid lipid nanoparticles, and nanostructured lipid carriers and polymeric, metallic, and hybrid nanoparticles are under pre-clinical investigation [163,164,165,166,167,168,169,170,171,172,173,174,175,176,177,178,179]. Despite this, there is still no nanomedicine commercially available for patients with melanoma so far, being limited to volunteers in the different clinical trials. Presently, liposomal formulations and polymeric nanoparticles are the most investigated nanosystems in clinical trials as reported in Table 5.

In the specific case of melanoma and to overcome the toxicity of paclitaxel in its free form, albumin-bound paclitaxel nanoparticles are one of the most investigated nanosystems in recent years. As an example, Hersh’s team conducted a phase 2 study (NCT00081042) that aimed to assess the safety and efficacy of ABI-007, an albumin-bound paclitaxel nanoparticle formulation, in the treatment of inoperable locally recurrent or metastatic melanoma. The study was divided into 2 cohorts of 37 patients each with or without previous chemotherapy, respectively. Overall, the formulation was well tolerated and demonstrated activity in both cohorts, with the median overall survival (OS) being 12.1 months for the chemotherapy-previously treated versus 9.6 months for the chemotherapy-naïve patients. Regarding this latter group, 8 patients discontinued treatment due to significant adverse effects such as neuropathy and myelosuppression [180].

Phase 2 clinical trial (NCT00404235) sought also to assess the safety and efficacy of albumin-bound paclitaxel nanoparticle but this time in combination with carboplatin. As in the previous clinical test, 76 patients with metastatic melanoma were divided into two cohorts depending on whether they had been previously treated with chemotherapy, a median OS of 10.9 and 11.1 months, respectively, was found. Furthermore, both cohorts observed some serious adverse effects such as neutropenia, leukopenia, fatigue, nausea and vomiting [181]. Later, the same sponsor promoted one more phase 2 study (NCT00626405) to analyze the safety and efficacy of chemotherapy (nanoparticle albumin-bound paclitaxel/carboplatin or temozolomide) and immunotherapy (bevacizumab) combined regimen. Although slightly more toxicity was observed in the bevacizumab-nanoparticle albumin-bound paclitaxel/carboplatin treatment group, efficacy results demonstrated a slight superiority of this, with a median OS of 13.9 months in comparison with the 12.3 months of the other cohort [182].

The comparison of the safety and efficacy of the nanoparticle albumin-bound paclitaxel/bevacizumab combination therapy regimen or the use of ipilimumab alone was investigated in another phase 2 clinical trial (NCT02158520). In the treatment group that integrated the nanoparticles, two complete and one partial response were found and serious adverse effects such as hemolytic uremic syndrome and rectal bleeding were observed. In contrast, only one partial response was achieved with immunotherapy as a single strategy. The majority of toxicities of this cohort were lymphocytopenia, benign or malignant neoplasms and acute kidney injury [183].

Furthermore, in a phase 2 study (NCT00462423), the safety and efficacy of the combination therapy of nanoparticle albumin-bound paclitaxel with avastin in 50 patients with metastatic melanoma were assessed. A median OS of 16.8 months was reached, and about 25% of patients developed some serious adverse effects. Pulmonary embolism, supraventricular tachycardia, macular edema, transient ischemic attack, nephrotic syndrome, renal tubular necrosis and dyspnea are examples [184].

Another example is related to a phase 3 study (NCT00864253) which aimed to compare the safety and efficacy of nanoparticle albumin-bound paclitaxel versus dacarbazine. A total of 529 patients with metastatic disease and chemotherapy-naïve patients were distributed into 2 groups. The nanoparticles demonstrated a clinical benefit compared to the use of dacarbazine (median OS 12.6 months versus 10.5, respectively) and a manageable safety profile. The most common adverse effects were neutropenia, leukopenia, alopecia and neuropathy [185].

One melanoma patient was included in the safety and pharmacokinetic and pharmacodynamic profile phase 1 study (NCT01300533) of another polymeric nanosystem containing docetaxel and targeting prostate-specific membrane antigen (PSMA). In terms of results, the formulation showed a different pharmacokinetic profile than the drug in the free form, with an increase in blood circulation time. Predictably and similarly to conventional docetaxel, the most common adverse effects included neutropenia, anemia, fatigue, alopecia, diarrhea and nausea [186].

In another example, seven volunteers with impaired liver function secondary to metastases participated in the characterization of the pharmacokinetic profile of a liposomal formulation of vincristine sulfate as well as in its safety and efficacy addressed in a single-arm phase 1 clinical trial (NCT00145041). A half-life of approximately 10 h, a clearance of 193 mL/h/m^2^, and a volume of distribution of 2722 mL/m^2^ were achieved. Moreover, blood, cardiac, gastrointestinal, respiratory and skin toxicities were detected [187]. The same principal investigator states, in another work [188], that the encapsulation of vincristine sulfate in liposomes resulted in a prolonged plasma half-life in comparison to the drug in its free form. The improved pharmacokinetic profile resulted in a greater accumulation of the compound at the tumor microenvironment, increased anticancer activity devoid of associated toxicity.

Finally, the safety and immunogenicity of escalating doses of a liposomal vaccine targeting dendritic cells, called Lipovaxin-MM, were also clinically evaluated in a phase 1 study (NCT01052142). Although most patients had disease progression, a partial response and disease stabilization in two other cases were found. Furthermore, the toxicological profile was quite favorable with more than 90% of adverse effects classified as grade 1 or 2 [189].

In addition, several other therapeutic strategies are now under clinical investigation and they are listed in Table 6. Others, such as the phase 1 clinical trial that pursues to investigate the safety and feasibility of a tumor mRNA-loaded liposomal vaccine will start in October 2022.

## 4. Conclusions

Nowadays, melanoma represents a serious public health problem. Due to its complexity and unpredictability, melanoma annually takes the lives of thousands of people around the world. Age, sex, ethnicity and individual phenotypic traits are examples of factors associated with increased risk. Nevertheless, melanoma is a highly preventable cancer since exposure to ultraviolet radiation is the principal risk factor. Thus, investment in the health literacy of the population is extremely important.

Additionally, melanoma is one of the cancers with the most complaints of pathology and dermatological malpractice due to misdiagnosis, which attests to the importance of early and accurate diagnosis and staging.

In terms of treatment, the current therapy for melanoma has reached a limit of clinical responses. Although the emergence of immuno and targeted therapies has resulted in an increase in the life expectancy of melanoma patients, some patients relapse or simply do not respond to these regimens. Therefore, the development of new and better therapies remains a priority for researchers, confirmed by the extensive number of ongoing preclinical and clinical trials. Herein, new drugs, combination therapies and nanotechnology-based strategies have been described which reveal a high potential for melanoma management. Overall, nanotechnological-based strategies show good histocompatibility, enhanced drug targeting, low toxicity and many other excellent characteristics, conferring broad application in the diagnosis and treatment of melanoma, particularly metastatic melanoma.

## Figures and Tables

**Figure 1 cancers-14-04652-f001:**
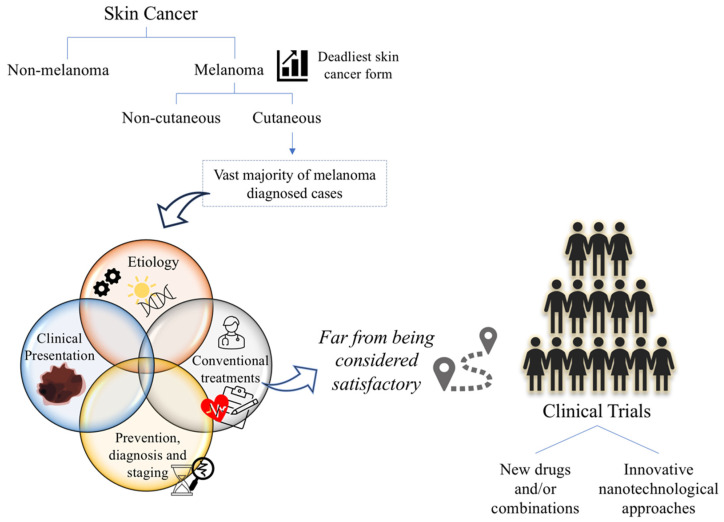
Melanoma: challenges and opportunities.

**Figure 2 cancers-14-04652-f002:**
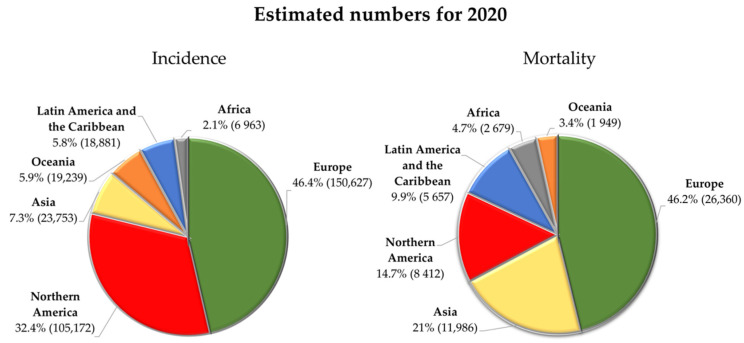
Estimated number of new skin melanoma cases and deaths worldwide in 2020 for both sexes and all ages. Data adapted from the Global Cancer Observatory by IARC [46,48].

**Figure 3 cancers-14-04652-f003:**
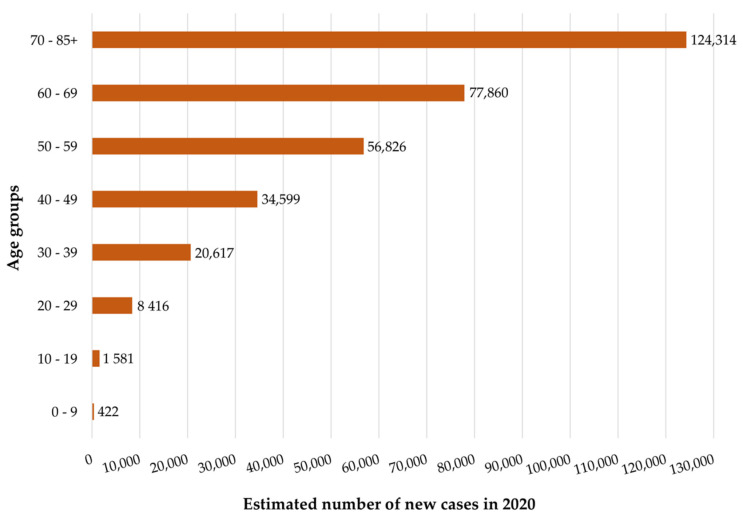
Estimated number of new skin melanoma cases worldwide for both sexes by age group. These data correspond to 2020 and were adapted from the Global Cancer Observatory by IARC [62].

**Figure 4 cancers-14-04652-f004:**
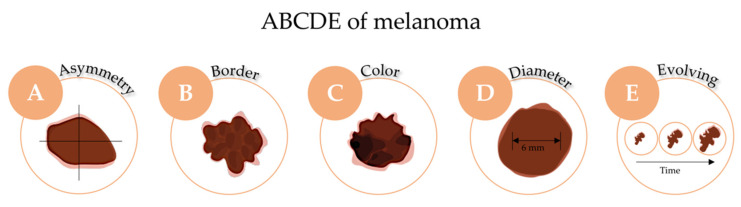
The ABCDE rule for melanoma skin cancer diagnosis.

**Table 2 cancers-14-04652-t002:** Melanoma treatment drugs and respective years of approval by Food and Drug Administration (FDA) and European Medicines Agency (EMA).

Type of Treatment	Mechanism	Drug	FDA Approval Date	EMA Approval Date
Chemotherapy				
	Alkylating agent	Dacarbazine	1975	2002
**Immunotherapy**				
	Antiviral	Interferon alpha-2b	1996	2000
		Peginterferon alpha-2b	2011	-
	Interleukin	Interleukin-2	1998	-
	Monoclonal antibody anti-CTLA4	Ipilimumab	2011	2011
	Monoclonal antibody anti-PD-1	Pembrolizumab	2014	2015
		Nivolumab	2014	2015
	Monoclonal antibody anti-PD-L1	Atezolizumab *	2020	-
	Monoclonal antibody anti-LAG-3	Relatlimab-rmbw *	2022	-
	Oncolytic herpes virus	Talimogene laherparepvec	2015	2015
**Targeted therapy**				
	BRAF inhibitor	Vemurafenib	2011	2012
		Dabrafenib	2013	2013
		Encorafenib *	2018	2018
	MEK inhibitor	Trametinib	2013	2014
		Cobimetinib *	2015	2015
		Binimetinib *	2018	2018

Abbreviations: BRAF: serine/threonine protein kinase B-raf; CTLA-4: T-lymphocyte-associated protein-4; LAG-3: lymphocyte activation gene-3; MEK: mitogen-activated protein kinase; PD-1: programmed cell death protein-1; PD-L1: programmed death-ligand 1. * Only used in combination with other medicines. Data collected from the database of each regulatory agency [112,113].

**Table 3 cancers-14-04652-t003:** Combination therapies for melanoma and years of approval by FDA and EMA.

Type of Treatment		Name of Drugs	FDA Approval Date	EMA Approval Date
**Combinatorial approaches**	**Targeted therapy**	Trametinib + Dabrafenib	2014	2015
		Cobimetinib + Vemurafenib	2015	2015
		Binimetinib + Encorafenib	2018	2018
	**Immunotherapy**	Nivolumab + Ipilimumab	2015	2016
		Nivolumab + Relatlimab-rmbw	2022	-
	**Targeted therapy + Immunotherapy**	Cobimetinib + Vemurafenib + Atezolizumab	2020	-

Data collected from the database of each regulatory agency [112,113].

**Table 4 cancers-14-04652-t004:** Examples of melanoma treatments undergoing clinical trials.

Clinical Trial Phase	Clinical Trial Description	Melanoma Stage	Sponsor	Estimated Starting or Completion Date	Trial ID
**1**	Safety and efficacy of the combination of ipilimumab and imatinib mesylate.	IV	M.D. Anderson Cancer Center	2013–2024	NCT01738139
	Safety of the combination of panobinostat (histone deacetylase inhibitor) and ipilimumab.	III/IV	H. Lee Moffitt Cancer Center and Research Institute	2014–2023	NCT02032810
	Safety and efficacy of the combination of imiquimod and pembrolizumab.	IIIB-IV	Mayo Clinic	2017–2023	NCT03276832
	Efficacy of intermittent dosing in the combination of vemurafenib and cobimetinib in the treatment of advanced BRAF V600 mutant melanoma with elevated levels of LDH.	IIIC-IV	H. Lee Moffitt Cancer Center and Research Institute	2018–2022	NCT03543969
	Safety of the administration of neoadjuvant atezolizumab treatment before surgery in non-metastatic resectable melanoma.	I/II	The Methodist Hospital Research Institute	2020–2025	NCT04020809
	Safety and efficacy of the combination of PeptiCRAd-1 and pembrolizumab.	Inoperable or metastatic	Valo Therapeutics Oy	2022–2024	NCT05492682
**2**	Efficacy of the combination of T-VEC and pembrolizumab.	III/IV	National Cancer Institute	2017–2023	NCT02965716
	Safety and effectiveness of the combination of PD-L1 (atezolizumab) and anti-VEGF (bevacizumab) therapies.	III/IV	Elizabeth Buchbinder	2020–2023	NCT04356729
	Efficacy of the combination of T-VEC and nivolumab.	IIIB/C/D/IVM1a	The Netherlands Cancer Institute	2020–2023	NCT04330430
	Safety and efficacy of the combination of pembrolizumab and infliximab.	III/IV	Massachusetts General Hospital	2022–2025	NCT05034536
	Safety and efficacy of the combination of PD-1 antibody tislelizumab and dacarbazine.	III/IV	Henan Cancer Hospital	2022–2024	NCT05466474
**3**	Efficacy of the immunization with natural dendritic cells as adjuvant treatment after complete radical lymph node dissection or sentinel node procedure.	IIIB/C	Radboud University Medical Center	2016–2024	NCT02993315
	Analysis of the safety, efficacy and pharmacokinetic between the combination of atezolizumab, cobimetinib and vemurafenib or combination of only cobimetinib and vemurafenib in previously untreated BRAF V600 mutant melanoma.	IIIC/IV	Hoffmann-La Roche	2017–2023	NCT02908672
	Assessment of fixed-dose combination of relatlimab and nivolumab versus nivolumab monotherapy after complete resection.	III/IV	Bristol-Myers Squibb	2021–2025	NCT05002569
	Safety and efficacy of the combination of encorafenib and binimetinib in comparison to placebo in BRAF V600E/K mutant melanoma.	IIB/C	Pierre Fabre Medicament	2022–2035	NCT05270044
**4**	Tolerability and long-term safety of dabrafenib and trametinib, alone or in combination.	Advanced or metastatic	Novartis Pharmaceuticals	2017–2027	NCT03340506
	Safety of pembrolizumab.	III/IV	Merck Sharp & Dohme LLC	2019–2026	NCT03715205

Abbreviations: BRAF: serine/threonine protein kinase B-raf; LDH: lactate dehydrogenase; PD-L1: programmed death-ligand 1; PeptiCRAd-1: peptide-coated conditionally replicating adenovirus-1; T-VEC: talimogene laherparepvec; VEGF: vascular endothelial growth factor. Data collected from the ClinicalTrials.gov database [126].

**Table 5 cancers-14-04652-t005:** Examples of completed clinical trials using different types of nanosystems for melanoma treatment.

Nanosystem	Main Clinical Trial Description	Melanoma Stage	Sponsor	Starting and Completion Date	Trial ID
Liposomes	Safety, efficacy and pharmacokinetic profile study of vincristine sulfate liposomes (Phase 1).	III/IV	Acrotech Biopharma LLC	2005–2007	NCT00145041
	Safety and immunogenicity of a dendritic cells targeted-liposomal vaccine (Phase 1).	IV	Lipotek Pty Ltd.	2009–2012	NCT01052142
Polymeric nanoparticles	Safety and efficacy of nanoparticle albumin-bound paclitaxel (Phase 2).	Unresectable or metastatic	Jonsson Comprehensive Cancer Center	2004–2010	NCT00081042
	Safety and efficacy of the combination of nanoparticle albumin-bound paclitaxel with carboplatin (Phase 2).	IV	Alliance for Clinical Trials in Oncology	2006–2010	NCT00404235
	Safety and efficacy of the combination of nanoparticle albumin-bound paclitaxel with avastin (Phase 2).	III/IV	Lynn E. Spitler, MD	2007–2012	NCT00462423
	Comparison of the safety and efficacy of the combination of bevacizumab, carboplatin and nanoparticle albumin-bound paclitaxel with the combination of bevacizumab and temozolomide (Phase 2).	IV	Alliance for Clinical Trials in Oncology	2008–2012	NCT00626405
	Comparison of the safety and efficacy of nanoparticle albumin-bound paclitaxel versus dacarbazine (Phase 3).	IV	Celgene	2009–2014	NCT00864253
	Safety and pharmacokinetic and pharmacodynamic profile of PSMA-targeted PLA/PEG docetaxel nanoparticles (Phase 1).	Advanced or metastatic	BIND Therapeutics	2011–2016	NCT01300533
	Comparison of the safety and efficacy of the combination of nanoparticle albumin-bound paclitaxel with bevacizumab versus ipilimumab alone (Phase 2).	IV	Academic and Community Cancer Research United	2013–2019	NCT02158520

Abbreviations: PEG: polyethylene glycol; PLA: polylactic acid; PSMA: prostate-specific membrane antigen. Data collected from the ClinicalTrials.gov database [126].

**Table 6 cancers-14-04652-t006:** Examples of ongoing or programmed clinical trials using different types of nanosystems for melanoma treatment.

Nanosystem	Main Clinical Trial Description	Melanoma Stage	Sponsor	Estimated Starting or Completion Date	Trial ID
Liposomes	Safety and tolerability of a liposomal tetravalent RNA-drug products vaccine (Phase 1).	IIIB/C/IV	BioNTech SE	2015–2023	NCT02410733
	Safety and feasibility of a tumor mRNA-loaded liposomal vaccine (Phase 1).	IIB-IV	University of Florida	2022–2027	NCT05264974
Lipid nanoparticles	Safety and efficacy of a lipid nanoparticle encapsulating mRNAs encoding a human T-cell co-stimulator and pro-inflammatory cytokines as monotherapy or in combination with durvalumab (Phase 1).	Advanced or metastatic	ModernaTX, Inc.	2018–2023	NCT03739931
Polymeric nanoparticles	Safety and efficacy of the combination of nanoparticle albumin-bound paclitaxel with bevacizumab (Phase 1).	IV	Mayo Clinic	2014–2025	NCT02020707
	Comparison of the safety and efficacy of the combination of nanoparticle albumin-bound paclitaxel and carboplatin with and without endostatin (Phase 2).	Advanced	Peking University Cancer Hospital & Institute	2019–2022	NCT03917069

Data collected from the ClinicalTrials.gov database [126].

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
