# Peer review of "Melanoma Management: From Epidemiology to Treatment and Latest Advances"

_cancers, 2022, doi:10.3390/cancers14194652_

Round 1

Reviewer 1 Report

Well written manuscript with important updates in the field, has my approval for a publication.

Author Response

Comments to the authors

Well written manuscript with important updates in the field, has my approval for a publication.

Reply: The authors would like to express their contentment that you liked our article and accepted it for publication.

Reviewer 2 Report

Melanoma is an important issue in cancer-related research. In this manuscript, the authors gave a comprehensive review of cutaneous melanoma, including various aspects from epidemiology to treatment. They also reviewed the latest advances in clinical trials, as well as nanotechnology-based strategies. The manuscript is well written with clear structure rich contents. Thus, I would recommend this work for publication in present form.

Author Response

Comments to the authors

Melanoma is an important issue in cancer-related research. In this manuscript, the authors gave a comprehensive review of cutaneous melanoma, including various aspects from epidemiology to treatment. They also reviewed the latest advances in clinical trials, as well as nanotechnology-based strategies. The manuscript is well written with clear structure rich contents. Thus, I would recommend this work for publication in present form.

Reply: The authors would like to express their satisfaction for having liked and accepted the manuscript for publication.

Reviewer 3 Report

In this review the authors present an updated overview concerning various aspects related to cutaneous melanoma and conventional treatments currently available, as well as, the latest advances in clinical trials regarding new drugs, including nanotechnology-based strategies. It is a very interesting and helpful text and along with the figures and the tables presented provides all the information needed as far as the subject. The way it is written is easy for the reader to follow and demonstrates the effort the authors made to complete it. Overall, the manuscript is of great significance and highlights that way the importance of the info presented.

Author Response

Comments to the authors

In this review the authors present an updated overview concerning various aspects related to cutaneous melanoma and conventional treatments currently available, as well as, the latest advances in clinical trials regarding new drugs, including nanotechnology-based strategies. It is a very interesting and helpful text and along with the figures and the tables presented provides all the information needed as far as the subject. The way it is written is easy for the reader to follow and demonstrates the effort the authors made to complete it. Overall, the manuscript is of great significance and highlights that way the importance of the info presented.

Reply: The authors would like to express their satisfaction  that you liked and accepted the manuscript for publication.

Reviewer 4 Report

In this manuscript authors described melanoma features and take into account the type,classification, eziology, diagnostic and possible treatments available. They also described the status of clinical trials on melanoma considering active studies. The review is well written and bibliography is updated. I suggest publication on Cancer journal after minor revisions:

-Check the entire paper for typos.

-I found the lines 495-498 not well related to the paragraph 3.2, I suggest to modify or better introduced the nanotechnolgy-related strategies or to move this part into the paragraph 3.3. 

-Lines 569-575: I suggest to specify with some data why the results obtained for liposomal formulation are better than the treatment with free drugs.

Author Response

Comments to the authors

In this manuscript authors described melanoma features and take into account the type, classification, etiology, diagnostic and possible treatments available. They also described the status of clinical trials on melanoma considering active studies. The review is well written and bibliography is updated. I suggest publication on Cancer journal after minor revisions:

The authors would like to express their profound gratitude to the reviewer for the constructive suggestions made to our article. We have carefully considered all comments, to which we have elaborated a point-by-point response. In the paper, all revised sections are highlighted in yellow.

Questions to the authors

  1. Check the entire paper for typos.

Reply: Thank you for your comment. The entire paper was revised and typos were rectified. Some self-corrections were also made.

  1. I found the lines 495-498 not well related to the paragraph 3.2, I suggest to modify or better introduced the nanotechnolgy-related strategies or to move this part into the paragraph 3.3.

Reply: Thanks for your suggestion. We appreciate and agree with referee’s point. We moved the referred part to the beginning of section  3.3.

  1. Lines 569-575: I suggest to specify with some data why the results obtained for liposomal formulation are better than the treatment with free drugs.

Reply: Thank you for your suggestion. In this specific clinical trial (NCT00145041), the authors formed a single group of 7 volunteers to evaluate the pharmacokinetic profile of a liposomal formulation of vincristine sulfate. As detailed now in the text, this clinical trial was carried out in a special melanoma patients population: with hepatic dysfunction secondary to liver metastases. A comparative arm for free drug administration was not included in this clinical trial. Moreover, to the best of our knowledge, and although the objectives of the clinical trial also included the assessment of treatment efficacy, these results have not yet been made public. However, some research work  of the same authors was consulted and reasons for the added value of this liposomal formulation of vincristine sulfate compared to the free drug were included in the revised version of the manuscript.

Once more, the authors thank Reviewer 4 for her/his valuable contribution to our article.

Reviewer 5 Report

The authors summarize the current state of melanoma detection and treatment, extending into ongoing and upcoming clinical trials.  I found the piece to be well organized and well written, with only a handful of typos I would suggest changing.

First, in several places "gender" is used interchangeably with "sex".  Since "sex" is specifically the non-modifiable risk factor intended, I suggest replacing mentions of "gender" with "sex". This includes Table 1.

Line 257 suggest changing "gets" with "becomes".

Line 468 suggest changing "associated to" to "associated with".

Author Response

Comments to the authors

The authors summarize the current state of melanoma detection and treatment, extending into ongoing and upcoming clinical trials.  I found the piece to be well organized and well written, with only a handful of typos I would suggest changing.

The authors would like to express their profound gratitude to the reviewer for the constructive suggestions made to our article. We have carefully considered all comments, to which we have elaborated a point-by-point response. In the paper, all revised sections are highlighted in yellow.

Questions to the authors

  1. First, in several places "gender" is used interchangeably with "sex". Since "sex" is specifically the non-modifiable risk factor intended, I suggest replacing mentions of "gender" with "sex". This includes Table 1.

Reply: Thank you for your comment. The proposed changes were performed in the revised version of the manuscript.

  1. Line 257 suggest changing "gets" with "becomes".

Reply: Thank you for your comment. The alteration was made.

  1. Line 468 suggest changing "associated to" to "associated with".

Reply: Thank you for your suggestion. The alteration was made.

Once more, the authors thank Reviewer 5 for her/his valuable contribution to our article.